# Brief Communication: Drivers of the recent warming of the Mediterranean Sea, and its implications for hail risk

Stephen Cusack[1], Tyler Cox[2]

[1]Stormwise Ltd, Luton, LU4 9DU, United Kingdom
[2]Inigo Limited, London, EC3A 5AY, United Kingdom

*Correspondence to*: Stephen Cusack (stephen.cusack@stormwise.co.uk)

**Abstract**

The Mediterranean Sea has warmed by about 2 K since the early 1980s, and this heating intensifies damage from moisture-driven perils such as hailstorms and floods in continental Europe via the basic thermodynamic effect. This study uses the DAMIP (Detection and Attribution Model Intercomparison Project) set of climate model experiments to explore the drivers of the recent Mediterranean warming. The models simulate the observed multidecadal variations of Mediterranean Sea temperatures in the modern period accurately, and indicate anthropogenic aerosol forcing was largely responsible for the cooler period from about 1900 to the late 1970s, while rising greenhouse gases are the main cause of warming waters since then. Next, we reviewed trends in damaging hail, and found they have been increasing by around 2% per year in key parts of mainland Europe since 1980. The rising risk fits with the established mechanism whereby warmer waters moisten low-level air leading to more severe thunderstorm hazards, though the exact relation between a warming Mediterranean and hail loss growth remains uncertain. Anthropogenic forcings will continue warming the Mediterranean for the next couple of decades at least, suggesting further increases to hail damages in high-risk parts of Europe.

## 1 Introduction

Climate impacts from anthropogenic forcings have regional variations, caused by factors such as local dynamical feedbacks and spatial inhomogeneities of some drivers such as aerosols (e.g. Seneviratne et al., 2021). The Mediterranean region has been a notable climate hotspot over the past few decades (e.g. MedECC, 2020; Ali et al., 2022). For example, sea surface temperatures warmed by 0.41 K/decade over the 1982-2023 period according to the Copernicus Marine Service (Roquet et al., 2016; Mulet et al., 2018) which is almost double the rate of the global oceans. Further, its warming is projected to continue outpacing the global mean in the future (e.g. Lionello and Scarascia, 2018).

One of the main effects of a warming sea is the humidification of the lower levels of the atmosphere. The warming sea surface is accompanied by similar increases in temperature of the overlying air, increasing its water-holding capacity, and as a result

more water is evaporated into the atmosphere. In general, greater amounts of low-level water vapour in the atmosphere act to raise the severity of weather perils such as heavy precipitation events (reviewed in Seneviratne et al., 2021) and large hail (e.g. Raupach et al., 2021; Chen and Dai, 2023). More specifically to central and southern parts of mainland Europe, moisture from the Mediterranean Sea was a key ingredient in many of the most extreme floods (e.g. James et al., 2004; Volosciuk et al., 2016; Krug et al., 2022; Tradowsky et al., 2023) and hailstorms ((e.g. Heimann and Kurz, 1985; Kunz et al., 2018; Piper et al., 2019; Kunz et al., 2020; Kopp et al., 2023) in recent decades.

Given how research has established low-level Mediterranean air masses as an important component of past major weather disasters, and that these air masses are moistening due to sea surface heating, it would be useful to learn more about the Mediterranean Sea warming trend, and its consequences on severe weather risk. The twin aims of this study are to gain insights into the drivers of the warming Mediterranean in recent decades, and to measure the trends in damaging hail over the same period. Section 2 contains a description of the data and processing used in this study, then in section 3 we analyse model simulations and identify the key drivers of multidecadal variations in Mediterranean temperatures over the industrial period. Section 4 begins with a review of how Mediterranean Sea temperatures affect the occurrence of damaging hail over the higher-risk parts of Europe, then we present estimates of trends in damaging hail over recent decades from previous studies, and a new analysis of insurance loss data. Conclusions are presented in Section 5.

## 2 Data and methods

### 2.1 Mediterranean temperatures

This study uses data from climate model experiments performed in the DAMIP (Detection and Attribution Model Intercomparison Project; Gillett et al., 2016) sub-project of the sixth version of the Coupled Model Intercomparison Project (CMIP6; Eyring et al., 2016). DAMIP was designed to investigate the impacts of various external forcings on global and regional climate. Its experiments explore the modern industrial period from 1850 to 2014, and its Tier 1 model simulations consist of setting one type of forcing to suitable values for the historical period, with all others fixed at pre-industrial values. Initial conditions are taken from pre-industrial control runs (from the main CMIP6 model experiments), and Tier 1 historical forcings are split into three distinct types: natural forcings (solar and volcanic, hereafter Nat), greenhouse gases (GHG), and anthropogenic aerosols (Aero). We analyse results from six different modelling centres providing monthly-mean near-surface temperature diagnostics (variable 'tas') for at least five different ensemble members, for all three forcing tests. In addition, the Historical experiments with all forcings (Hist) performed as part of the central CMIP project are also analysed. Table 1 summarises the simulations analysed in this study. More information on the climate models is provided in CMIP (2025).

Observed sea surface temperatures (SSTs) are taken from the Hadley Centre Sea Ice and Sea Surface Temperature data set (HadISST; Rayner et al., 2003), a global dataset of monthly sea surface temperature values at 1° x 1° resolution from January 1870 and continually updated to the present day. These SST data are used to assess the validity of near-surface air temperature variations in DAMIP simulations. Anomalies in both quantities are very similar at the large spatial and temporal scales analysed

in this study. For example, Rubino et al. (2020) described their close correspondence at interdecadal timescales (their Figure 4), and the two quantities are often used interchangeably in long climate reconstructions (e.g. Morice et al., 2012).

65    **Table 1: summary details of the DAMIP climate model simulations.**

| Model | Reference ID in text | No. of ensemble members | Release year | Atmosphere resolution (km) | Ocean resolution (km) |
|---|---|---|---|---|---|
| CNRM-CM6-1 | 1 | 10 | 2017 | 250 | 100 |
| CanESM5 | 2 | 15 | 2019 | 500 | 100 |
| GISS-E2-1-G | 3 | 5 | 2019 | 250 | 100 |
| HadGEM3-GC31-LL | 4 | 15 | 2016 | 250 | 100 |
| MIROC6 | 5 | 10 | 2017 | 250 | 100 |
| MPI-ESM1-2-LR | 6 | 15 | 2017 | 250 | 250 |

Both observed and modelled data are processed similarly. Monthly mean values are initially area-averaged over a region of the Mediterranean Sea shown in Figure 1, then combined to form an extended summer half-year average (May-October)
70    corresponding to the annual peak of Mediterranean influence on flood and hail risk, in order to focus on the most relevant seasonal warming trends (García-Monteiro et al., 2022). Anomalies were defined using the climate from the common baseline period of 1870-2014 for observed and modelled values. The climate of the Hist simulations in this common 1870-2014 period has been used to define all model anomalies, for consistency with how observed anomalies are defined. Finally, a second-order low-pass Butterworth filter (Butterworth, 1930) with a 5-year cutoff was applied to all timeseries to reduce large amounts of
75    interannual noise obscuring longer-term trends. The 5-year cutoff was chosen to retain potential signals from Nat forcings such as 11-year solar cycles and major volcanoes, as well as the slower changes from GHG and Aero.

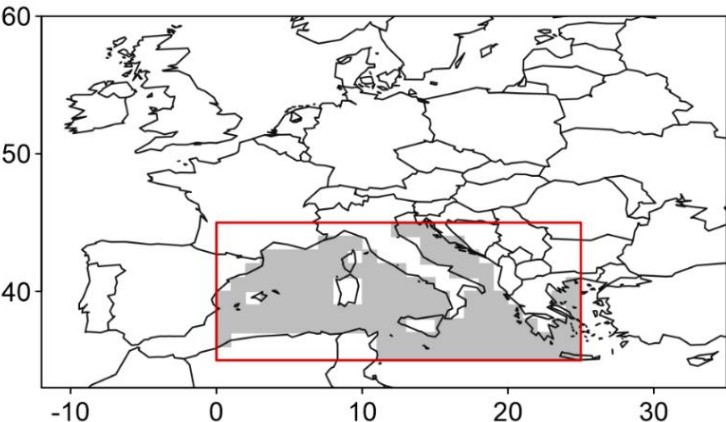

**Figure 1: map of a part of Europe, with shaded area denoting the Mediterranean Sea region used in the analysis.**

## 2.2 European hail losses

National insurance associations often monitor losses to their market, and some of their records extend to the past few decades. Three publicly available historical records of hail losses are explored in Section 4 for guidance on trends in the risk from this weather peril. These data are now described.

The German Insurance Association (GDV) issue annual losses indexed to 2023 inventory and prices using data on numbers of insured vehicles, and the cost of automobile parts and repair from the Federal Statistical Office of Germany (GDV, 2024).

The second dataset comprises annual hail losses to all insured buildings in France, based on information from France Assureurs (2023). Their annual data consists of the ratio of all insured risks making a hail claim (a frequency ratio) and the average hail claim size indexed to 2022 values using the standard FFB Index, which is based on costs for various elements of a standard apartment in Paris and specifically designed to index insurance policies (FFB, 2025). We have computed total industry losses by multiplying these two quantities together, then scaling them by the total number of industry risks (N). The total industry loss in 2022 was used to define N, then we apply this value of N to all other years to obtain annual industry losses for all years based on the number of risks in 2022. This approach ensures indexation captures both the growth in claim severity, and how claim numbers increase with changes in number of insured risks.

The third national loss dataset contains annual hail loss costs in Switzerland for the 1980 to 2023 period for all buildings in the 19 Swiss cantons covered by public insurance, from VKG (2022, 2024). The loss cost is defined as the ratio of the total cost of repair to the total exposure value. It is a useful loss metric in insurance, because total exposure value in the denominator is usually defined to reflect the same growths in insurance claim size and number which are present in the numerator, hence socio-economic trends are absent from the ratio: loss costs are indexed (aka normalised) by design.

Trends in losses over the past few decades are derived from a regression of log(loss quantity) with year, providing best-fit estimates of the growth in units of % per year.

## 3 Results

### 3.1 Past changes in Mediterranean SST

Figure 2 displays timeseries of SST anomalies in both the global (60°S to 60°N) and Mediterranean regions for summer half-years in 1940 to 2024. The Mediterranean region cooled from roughly 1950 to 1980, then warmed rapidly since then. Changes in the global mean have been more muted, with anomalies near zero from 1940 to the mid-1970s, and a more gradual warming since then. Figure 2 also contains the best-fitting linear trend for SST in these two areas from 1980 to 2024, and it was found that average SSTs over each area have been warming at rates which are significantly different from zero at the 1% level (p-values are below 1e-10 in both cases). However, the most notable feature is the summertime Mediterranean SSTs trending

upward at more than three times the global-mean warming rate since 1980. The Mediterranean Sea has been an oceanic hotspot over the past four decades.

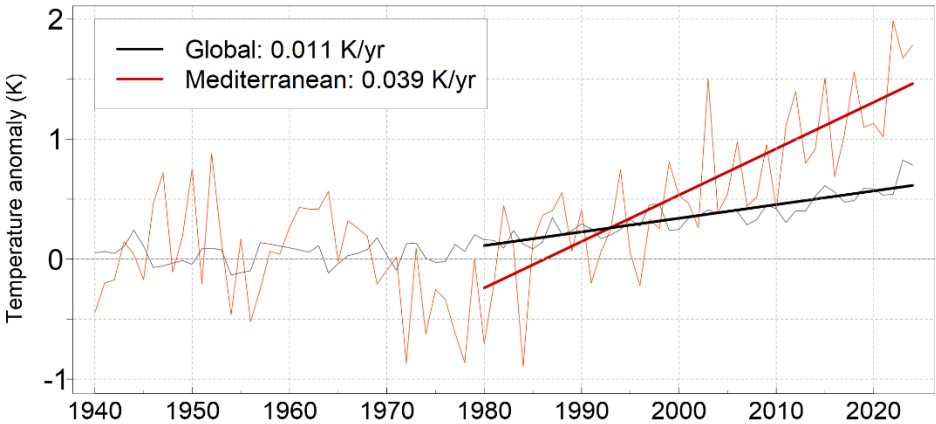

**Figure 2: timeseries of global (black) and Mediterranean (red) SST anomalies in the summer half-year (May to October), with associated best-fitting linear trends in the 1980-2024 period. Data are from HadISST, anomalies are computed with respect to the 1870 to 2014 period, and global data are from 60°S to 60°N.**

## 3.2 Drivers of past changes in DAMIP experiments

The causes of the more rapid warming of the Mediterranean Sea are now explored using DAMIP modelling results. Figure 3a shows the timeseries of Mediterranean temperature anomalies over the extended historical period for observed and multi-model ensemble means for the Historical and three DAMIP single-forcing experiments. Earlier multidecadal variations in observations, consisting of minima around 1910 and the late 1970s, and a local maximum from about 1930 to the early 1960s, are replicated in Hist, albeit with reduced amplitude. To our knowledge, there are no published analyses on the relative roles
of internal climate variability and external forcing toward the observed multidecadal variations in the Mediterranean basin prior to 1980. However, these earlier fluctuations are similar to those in the North Atlantic sector analysed by Booth et al. (2012), which they found to be caused by both anthropogenic and volcanic aerosol forcing, in addition to internal climate variability. The observed warming since about 1980 is of greater concern since it has four times larger amplitude than the mid-20[th] century peak. Notably, the multimodel ensemble mean simulates this recent rapid warming accurately. This comparison
to observed variations bolsters confidence in the fidelity of CMIP6-DAMIP climate model simulations of surface temperature anomalies in the Mediterranean, particularly its dominant feature of rapid warming in recent decades. We now use the validated model ensemble to assess the contribution of individual forcings to the total signal.

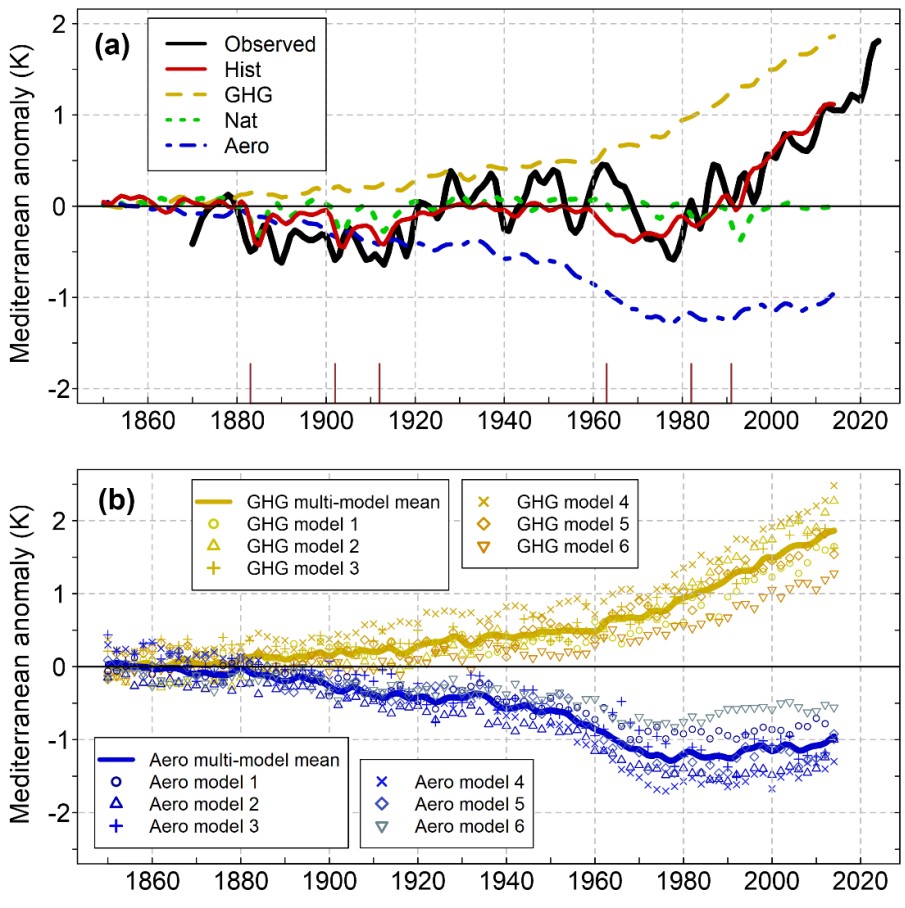

**Figure 3: Timeseries of Mediterranean temperature anomalies. (a) observed (solid black line) and multi-model mean anomalies for various forcing experiments, with major volcanic eruptions indicated by the long red tick marks on the time axis. (b) DAMIP multi-model mean (solid lines), and the mean of each of the six models (various symbols), for the Aero (blue) and GHG (gold) experiments.**

The impact of Nat forcing (green dotted line in Figure 3a) on Mediterranean temperatures is simulated to be near-zero over the long-term. However, volcanic eruptions exert influence on temperature anomalies at timescales shorter than about 10 years in DAMIP simulations. Six of the largest climate-changing eruptions in the simulation period are indicated as long red tick-marks on the x-axis (Krakatoa 1883; Santa Maria 1902; Novarupta 1912; Mount Agung 1963; El Chichon 1982; Pinatubo 1991), and are associated with anomalies of up to -0.5 K for a few years following the eruption. Such a size of perturbation is

smaller than the recent GHG and Aero forcings, and of much shorter duration, hence DAMIP model results indicate Nat forcing is a minor consideration in the modern industrial period.

GHG (gold dashed line) acts to warm the Mediterranean throughout the period. The magnitude of its forcing was small up to around 1920, and has grown continuously since then. DAMIP models indicate GHG had caused almost 2 K of warming by the end of the simulations in 2014, and caused almost three-quarters of the total modelled warming from 1975 to 2014.

The Mediterranean cooled in response to Aero forcing (blue dash-dotted line in Figure 3a). This forcing dominated total climate forcing up to 1920, and outweighed GHG forcing for much of the 20th century. Its peak impact on the Mediterranean was reached in the late 1970s, then stabilised for a short time before reducing in magnitude from about 1990 onwards, in line with aerosol emissions (e.g. Lund et al., 2019). The net warming from 1990 to 2014 due to reduced aerosol burdens caused almost one-quarter of the total modelled warming in this recent period.

Finally, we note how the sum of responses to individual forcings are consistent with Hist signals (not shown), suggesting Mediterranean warming is largely explained as a linear response to external forcings. Confidence in these DAMIP signals was also assessed using the inter-model spread. Figure 3b contains mean signals for each of the six climate models, for the two main forcings of GHG and Aero. All six models agree on the respective sign of the two forcings, and signal amplitudes are similar too. We conclude that GHG and Aero forcings of Mediterranean temperatures are a robust feature of DAMIP models

and that these two forcings are the primary drivers of recent Mediterranean Warming.

## 4 Trends in hail risk

### 4.1 Multidecadal connection from Mediterranean temperatures to hail risk

It is difficult to precisely measure the impacts that Mediterranean temperatures have on hail risk at multidecadal scales, mainly

due to the incompleteness of hail damage records both spatially and temporally, though their strong connection has been established. Basic thermodynamic principles indicate a warming sea surface is accompanied by similar increases in temperature of the overlying air, increasing its water-holding capacity, and as a result increasing the low-level moisture in the atmosphere. Further, the low-level moisture is an important ingredient for severe hail events. Indeed, previous studies of extreme damaging hail events in Europe reveal how their low-level moisture is drawn from the Mediterranean basin (e.g.

Heimann and Kurz, 1985; Kunz et al., 2018; Piper et al., 2019; Kunz et al., 2020; Kopp et al., 2023). Kunz et al. (2020) performed a more comprehensive analysis of hailstorms in their study area (Germany, France, Belgium, Luxembourg) and noted that the typical case involves a trough drawing moist Mediterranean air northwards over Europe on its eastern flank, while the increased shear and lifting processes from the trough produce conditions conducive for organised thunderstorm development. Other sources of low-level moisture may also cause damaging hail in some parts of Europe, such as the

Cantabrian Sea (e.g. de Pablo Dávila et al., 2021) and the Black Sea (e.g. Piper et al., 2019), though the Mediterranean is the largest and warmest body of water in the region, hence very likely the main source of the high dewpoints in the damaging hail events in the higher risk areas of central and western mainland Europe.

A number of studies have expanded on the connection between hail risk, low level moisture, and a warming Mediterranean to show that low-level moisture increases have been a main driver of trends in damaging hail in high-risk parts of Europe over the past few decades, via two pathways. The first pathway concerns moisture driving convective instability, hence greater amounts of water vapour will tend to intensify thunderstorms and produce more severe hail hazard and losses. A number of studies identify rising amounts of low-level moisture as the main cause of upward trends in damaging hail over the past few decades, in different parts of the higher-risk zone in central and western Europe: for example in France (Dessens, 1995), Germany (Kunz et al. 2009; Mohr and Kunz, 2013), Switzerland (Wilhelm et al., 2024), northern Italy (Battaglioli et al., 2023) and across the higher-risk areas of central and western Europe (e.g. Rädler et al., 2018). Further, multiple climate model projections suggest rising low-level moisture will continue to drive upward trends in damaging hail in the future (Rädler et al., 2019). The second pathway relating moisture to damaging hail concerns the rising frequency of extremely strong fronts. Schemm et al. (2017) found very strong fronts became more frequent over the bulk of Europe from 1979 to 2014 due to rising humidity, and Kunz et al. (2020) described how hail associated with fronts were more damaging, due to larger hailstones and longer hail swathes.

The occurrence of individual severe hail events depends on other quantities such as deep layer shear, steep mid-level lapse rates and the low-level cap, and any long-term changes to these quantities could also modulate hail risk. Past studies have investigated these other ingredients and find relatively steady, or even slightly inhibiting trends (e.g. Mohr and Kunz, 2013; Rädler et al., 2018; more fully documented in Taszarek et al., 2021). These results support increasing low-level moisture amounts as the main driver of upward trends in hail risk over the past few decades.

The full picture of trends in hail risk at all European locations will require consideration of other moisture source such as the Black Sea, and other environmental ingredients such as the warmer, drier air that inhibits convective initiation, and future research into their contributions would be valuable. Our current knowledge suggests it is very likely that a warming Mediterranean is the main contributor to the recent trends in damaging hail across the higher-risk parts of Europe. We now seek to quantify the trend in hail risk in the higher-risk areas, for guidance on how a warming Mediterranean may be impacting this peril.

**4.2 Measured trends in hail risk**

Specific measures of recent trends in hail damage fall into two broad types: those reflecting hailstone occurrence, and those based on observed hail damages. The former type has the benefit of more homogeneous hazard data over time, but are often based on models of hail occurrence at a location given the atmosphere conditions, with uncertainty as to whether the models accurately capture the relative roles of all hailstorm ingredients over the past few decades. In contrast, evaluations based on damages reflect hail occurrence at the ground, however, there is some uncertainty in estimated losses at the time of older events, and in how the losses are indexed to represent the total damages if the events occurred in the present day. Given their uncertainties, we will consider estimates of trends from both methods.

Hazard-based studies generally find that large hail has become increasingly common across most of Europe since the 1970s. For instance, extensive hailpad networks indicate a trend toward more intense hail in both southern France over the 1989-2009 period (Berthet et al., 2011), and the central-eastern Alpine area of Italy over the 1975-2009 period (Eccel et al., 2012). A second type of hazard-based study provides more spatially complete and longer hail reconstructions, based on weather ingredients from reanalyses and calibrated to observations such as damaging hail on the ground (e.g. Mohr and Kunz, 2013;

Rädler et al., 2018; Battaglioli et al., 2023). The hail model developed by Rädler et al. (2018) indicated the annual amount of hail ≥ 2 cm increased at around 2.4% p.a. over 1979-2015 in a region covering Germany and the Alps, and up by 1.4% p.a. over the greater area of western and central Europe. More recently, Battaglioli et al. (2023) described a model which had been calibrated to observed lightning and large hail reports, then estimated linear trends in damaging hail occurrence from 1950 to 2021, and found it had been rising generally across Europe, with more positive changes for larger hail. They highlighted how

hailstones ≥ 5 cm are now three times more likely than in the 1950s in northern Italy, though the trend was not uniform throughout the whole period, with most of the increase occurring since the 1980s, hence linear trends since 1950 are likely to underestimate trends over the past four decades. Some insurance industry research used models similar to those of Rädler et al. (2018) and Battaglioli et al. (2023): Stormwise Ltd. (2024) studied linear trends over 1960-2023 while Partner Re (2024) examined 1950-2022, and both report hazard driving increases in annual hail risk of around 1 to 1.5% p.a. over much of

mainland Europe. The trends from Rädler et al. (2018) are higher than others, and likely due to their trends representing the more recent 1979-2015 period hence more aligned with the period of Mediterranean warming, since about 1980.

Studies of property losses also show an upward trend in hail risk over recent decades. Many in the insurance industry get their impression of rising hail severity in Europe from publicly issued industry loss estimates. There have been some notable events in the past few years, with 4.5 billion USD of insured losses from severe thunderstorms in Europe during summer 2021 (Swiss

Re, 2021), then in 2022, French property and automobiles suffered five billion euros of insured loss due to hail (France Assureurs, 2023), while in 2023, hail storms caused over five billion euros of insured loss, mostly in Italy (Swiss Re, 2024). These loss severities were rarer in earlier times: storms Andreas (2013) and Ela (2014) caused multi-billion euro insured losses, mainly to Germany and France respectively (Swiss Re, 2014 and 2015) and were the first major events since the multi-billion loss in the Munich hailstorm of 1984 (e.g. Púčik et al., 2019). While the sudden spate of multi-billion euro events is impactful,

it provides relatively weak evidence of rising hail risk, due to significant errors in estimates of damages from older storms and statistical uncertainty from small sample sizes.

Consideration of other data sources and analysis provides a more robust view on hail loss trends. Early work included a study by Dessens (1995) which found an upward trend in crop-hail insurance losses in France after 1980, while Kunz et al. (2009) examined damage to buildings in southwest Germany over the period 1986-2004, and found the number of severe hail days

had increased significantly, by around 3.5% per year.

We supplement earlier damage studies with a new analysis of annual hail losses published by national insurance associations, which were described in subsection 2.2. These industry bodies often monitor losses to their market, and some of their records extend to the past few decades. Figures 4a-c show timeseries of hail losses in Germany, France and Switzerland respectively.

The timeseries of annual losses in the 1973-2023 period for automobiles in Germany due to the combined wind and hail perils (the vast majority of which are due to hail) contains an annual growth rate of 3.1% over the past 50 years, while more modern data since 1980 grow by 2.2% per year, and both trends are significantly different from zero at the 1% level based on a standard two-sided t-test (used in all significance testing of trends). Figure 4b presents annual hail losses to all insured buildings in France, containing a +5.8% annual growth rate of hail losses from 1990 to 2022, which drops to +4.7% p.a. when excluding the final year with its huge hail losses. Both of these trends are different from zero at the 1% level. Finally, Figure 4c shows annual hail loss costs in Switzerland for the 1980 to 2023 period for all buildings in the 19 Swiss cantons covered by public insurance, and the best-fitting trend is 2.6% per year over the past 44 years, and *not* different from zero at the 1% level (p-value = 0.076).

In summary, the datasets shown in Figure 4a-c indicate losses have been trending upward at 2 to 5% per year over the past few decades, and consistently faster than the 1 to 2% range suggested by the above-mentioned hazard-based studies. Losses from national bodies are based on accurate information from surveys of reported losses combined with careful indexation methods using data widely regarded as suitable for its purpose, and represent the industry's best estimates of loss growth. However, hazard-based estimates use hail models based on our best understanding of atmospheric processes, and calibrated to observed hail on the ground, and they represent the best estimates from meteorology. The cause of their different growth rates is unclear, and future investigation into how fast hail damage has been increasing would be useful. In the meantime, we suggest the best estimate of rising hail risk to property is around 2% per year for mainland Europe.

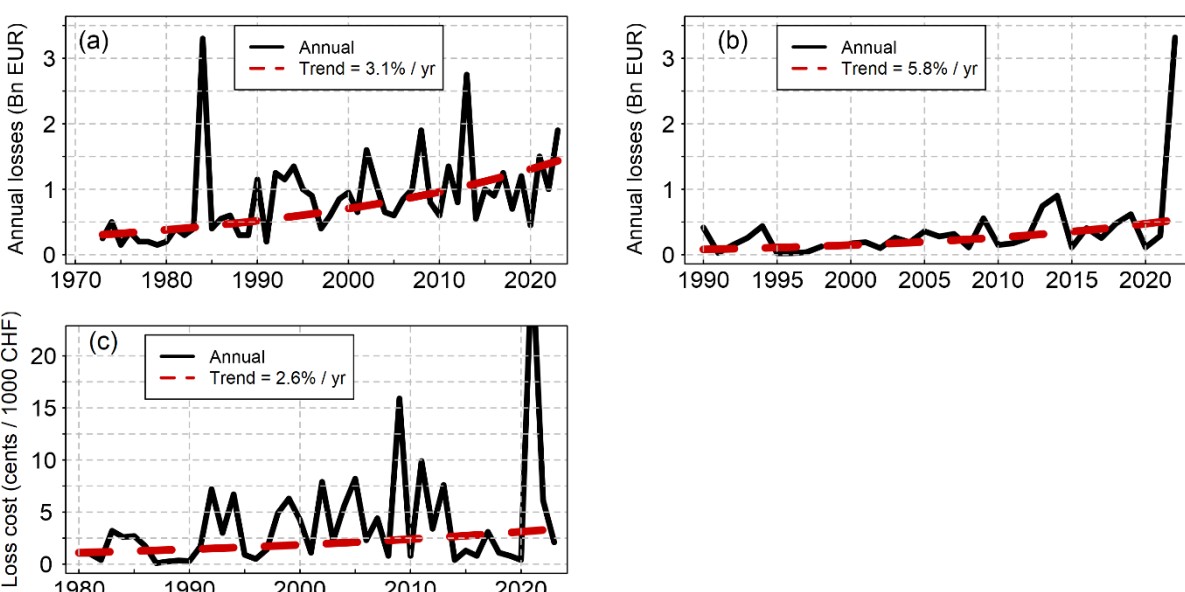

**Figure 4: timeseries of national hail damages for three countries. Insurance industry hailstorm losses are shown for (a) automobiles in Germany (GDV, 2024), (b) buildings in France (France Assureurs, 2023), and (c) buildings in Switzerland (VKG, 2022, 2024).**


In practice, a 2% per year trend in hail risk may accumulate to harm insurance companies. Their view of risk is often calibrated to past experience hence may lag behind current levels of risk. For example, if a view of risk is calibrated to a time period with a midpoint of 10 years ago, and the risk has been increasing at 2% per year, then this translates to a $1.02^{10} = 21.9\%$ underestimate of the present-day risk. This is material in the context of larger companies typically aiming for annual profits of
around 5 to 10%.

## 5 Conclusions

The Mediterranean Sea has been warming at three times the rate of global mean SSTs over the past 45 years (Figure 2), and drivers of this rapid heating were investigated using results from the CMIP6-DAMIP experiments. DAMIP climate models were found to reproduce the observed variations in observed Mediterranean temperatures at multidecadal timescales since
1870, especially the rapid heating in recent decades, building confidence in the fidelity of these multimodel mean estimates. They indicate anthropogenic forcings from greenhouse gases (GHG) and aerosols (Aero) have had the greatest influence on multidecadal changes in Mediterranean temperatures. Anthropogenic aerosols dominated up to the late 1970s to produce a slightly cool period on average, then clean-air acts reduced their impacts while GHG forcing continued to increase, resulting in a rapid warming of the Mediterranean of about 0.4 K / decade since 1980. Rising GHG amounts were responsible for about
75% of this warming over the past four decades, with the remaining portion mostly due to declining aerosol burdens.

The Mediterranean Sea is the main source of the high moisture amounts that are a key ingredient of the damaging hailstorms in the higher risk parts of Europe. We synthesized various studies and datasets to measure trends in European hail risk since 1980. Two different types of studies produce a similar sign of trend, but different rates of growth in risk over recent decades. First, models of hail based on environmental conditions and calibrated to observed occurrences of larger hail indicate damage
across mainland Europe has been rising by 1 to 2% per year since the mid-20$^{th}$ century due to ongoing warming, and the trend since 1980 is nearer the upper end of this range. Second, trends in estimates of hail damage to property, indexed to the present-day, show annual hailstorm losses are increasing by 2 to 5% in some key countries (Germany, France and Switzerland) above exposure-driven growth. Both methods have strengths and weaknesses, and future work on resolving their different estimates would be useful. We suggest a best estimate of 2% per year growth in hail damage based on current evidence. Uncertainty is
large, though large-scale trends in the higher-risk areas of western and central Europe are positive from almost all studies.

Impacts of rising trends in damaging hail are material to society. For example, if an estimate of hail risk was based on a climate centred on 10 years ago, then a 2% per year growth accumulates to a 22% underestimate of the damages from a peril which causes billions of euros of losses to Europe each year.

Past research establishes how the warming Mediterranean played a dominant role in upward trends of damaging hail over the
past few decades, though further work is required to precisely measure the relation. In this study, we find that the current

trajectory of anthropogenic forcings indicate sustained warming of the Mediterranean, and we recommend including upward trends in views of hail losses across high-risk parts of mainland Europe.

**Acknowledgements**

The authors are grateful to the reviewers and the public commentator for feedback which improved this manuscript, and to Inigo Limited for funding part of this research.

**Code/Data availability**

HadISST data are available at https://www.metoffice.gov.uk/hadobs/hadisst/ (accessed on 24th October 2024). CMIP6-DAMIP climate model results were downloaded from the Earth System Grid Federation (ESGF): https://aims2.llnl.gov/search/cmip6/
(accessed July-October 2024).

**Author Contribution**

SC designed the tests and did the analysis. SC and TC prepared the manuscript.

**Competing Interests**

The authors declare no competing interest.

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
