# Peer review of "Brief Communication: Drivers of the recent warming of the Mediterranean Sea, and its implications for hail risk"

_Natural Hazards and Earth System Sciences, 2024_

## Referee Comment (RC3)

**Review of NHESS-2024-210**

This study claims to investigate trends in European hailstorm damage. The study presents two components. First, insurance data are used to show increasing insured losses owing to hail over recent decades. Yet there is no analysis of exposure or vulnerability which are likely to be significant factors in changes in insured losses. Second, the paper shows an analysis of warming of northern Mediterranean waters, and shows that anthropogenic greenhouse gas emissions are driving these temperature increases. Yet, the link between the water temperature in the northern Mediterranean Sea and hailstorms is barely discussed, with only one line in Section 1 claiming that the "key area" is the Mediterranean Sea. The authors go on to make statements such as "given the coupling between northern Med temperatures and thunderstorm activity" and "a continuation of the warming trend in the Med corresponds to further increases in European hail damage" without sufficient evidence.

While the paper is well written and admirably concise, the article shows incomplete trend analysis and lacks a clear and properly justified link between the Mediterranean Sea temperature and hail damage. Since correlation is not causation, much more analysis is required to show the impact of rising Mediterranean Sea temperatures on hail damage in Europe. Otherwise, the article shows only that greenhouse gas emissions are driving rising Mediterranean Sea temperatures – since there is already an extensive body of work on climate change effects on the Mediterranean Sea (e.g. Ali et al., 2022), is this result novel?

I have made some specific suggestions below, but overall I would suggest that the authors a) consider changes in exposure and vulnerability in concert with changes in damages when performing trend analysis, and b) thoroughly link rising Mediterranean Sea temperatures to hailstorm activity (specifically, as compared to overall thunderstorm activity) in Europe.

**Specific comments**

1. Lines 15-25: In the introduction the authors should be more specific – what is meant by "recent times"? Were the June 2021 losses caused by hail or by flood as hinted at by the authors mentioning saturated soils? What is "similar magnitude" in the case of the Munich storm? "A couple of decades" should be replaced by the actual time period examined.

2. Lines 75-76: The question with insurance loss data is always the proportion of the trend

owing to change in the number or type of insured objects (ie the exposure and vulnerability part of the risk). See for example Strader et al. (2024) for tornado risk. The authors should comment on this aspect of uncertainty, and whether they think it plays a big role in the trends they present here.

3. Line 89: "Past research points to a warming climate causing more damaging hail in Europe". Yes, but the uncertainty must always be addressed because there remain many unknowns, as the authors have mentioned previously with e.g. references to Raupach et al. (2021). The authors should include the uncertainty in this statement.

4. Line 125: The Butterworth filter and its use requires a reference.

5. Line 187: "rising trend in the European hail climate" – this needs more specificity, since observed trends depend heavily on whether frequency or severity is considered, the geographical region, whether only hail of certain sizes is considered, etc.

6. Line 194: "Rising temperatures of local seas humidify the low-level air, which intensifies thunderstorms leading to more severe hail". The key word here is "local". How much effect would rising Mediterranean temperatures have on hailstorms far inland in Europe? This kind of link needs to be much further explored in this article.

7. Line ∼200: While aerosol changes have temperature effects, including on the Mediterranean as discussed here, they also have separate effects on thunderstorm activity that are highly uncertain and may exacerbate or modulate the changes owing to increased Mediterranean Sea temperatures. The authors should comment on this.

**Technical corrections and typos, etc**

1. Line 42: Remove repeated "more recently".

2. Line 83: Suggest removal of "more accurate" since CAPE and wet-bulb temperatures measure different things.

**References**

Ali, E., W. Cramer, J. Carnicer, E. Georgopoulou, N. Hilmi, G. L. Cozannet, and P. Lionello, 2022: Mediterranean region. *Climate Change 2022: Impacts, Adaptation and Vulnerability. Contribution of Working Group II to the Sixth Assessment Report of the Intergovernmental Panel on Climate Change*, H. O. Pörtner, D. C. Roberts, M. Tignor, E. S. Poloczanska, K. Mintenbeck, A. Alegría, M. Craig, S. Langsdorf, S. Löschke, V. Möller, A. Okem, and B. Rama, Eds., Cambridge University Press, Cambridge, UK and New York, NY, USA, doi: 10.1017/9781009325844.021.

Raupach, T. H., and Coauthors, 2021: The effects of climate change on hailstorms. *Nat Rev Earth Environ*, **2 (3)**, 213–226, doi:10.1038/s43017-020-00133-9.

Strader, S. M., V. A. Gensini, W. S. Ashley, and A. N. Wagner, 2024: Changes in tornado risk and societal vulnerability leading to greater tornado impact potential. *npj Natural Hazards*, **1 (1)**, 20, doi:10.1038/s44304-024-00019-6, URL https://doi.org/10.1038/s44304-024-00019-6.

---

## Author Comment (AC1)

**Reviewer 3**

The authors thank Reviewer 3 for their time providing feedback which improved the manuscript.

*This study claims to investigate trends in European hailstorm damage. The study presents two components. First, insurance data are used to show increasing insured losses owing to hail over recent decades. Yet there is no analysis of exposure or vulnerability which are likely to be significant factors in changes in insured losses.*

Thanks for this comment. It indicates how the description of indexation in the original manuscript was insufficient. The description of loss data and indexation to a common year has been moved to section 2.2 in the revised manuscript. These losses have been indexed by industry experts to take account of the factors found to be significant for insured losses. The revised manuscript (lines 86-100) provides more details on how these national industry bodies have indexed their losses to take account of evolving claims.

*Second, the paper shows an analysis of warming of northern Mediterranean waters, and shows that anthropogenic greenhouse gas emissions are driving these temperature increases. Yet, the link between the water temperature in the northern Mediterranean Sea and hailstorms is barely discussed, with only one line in Section 1 claiming that the "key area" is the Mediterranean Sea. The authors go on to make statements such as "given the coupling between northern Med temperatures and thunderstorm activity" and "a continuation of the warming trend in the Med corresponds to further increases in European hail damage" without sufficient evidence.*

The revised manuscript presents more evidence linking Mediterranean Sea temperatures to hailstorm damage in Europe. The Introduction contains an overview, with a description of how warming waters humidify the air in lines 27-29 of the Introduction, then followed by the association of Mediterranean Sea temperatures with hailstorms in lines 31-34. Section 4 of the revised text gives fuller details on the connection from Mediterranean moisture to hail trends, in lines 171-193, including several studies showing a direct influence of the Mediterranean Sea moistening low-level air in past hailstorms. We trust the revised manuscript conveys how past research results have established a link from Mediterranean temperatures to hail damage.

*While the paper is well written and admirably concise, the article shows incomplete trend analysis and lacks a clear and properly justified link between the Mediterranean Sea temperature and hail damage. Since correlation is not causation, much more analysis is required to show the impact of rising Mediterranean Sea temperatures on hail damage in Europe. Otherwise, the article shows only that greenhouse gas emissions are driving rising Mediterranean Sea temperatures – since there is already an extensive body of work on climate change effects on the Mediterranean Sea (e.g. Ali et al., 2022), is this result novel?*

- We appreciate your feedback on the quality of writing.
- Both trend analysis, and the link from Mediterranean to hail damage, are discussed in the paragraphs above.
- To our knowledge, the use of DAMIP climate model results to quantify the relative contributions of external forcings on Mediterranean temperatures is new. For instance, Ali et al. (2022) describe temperature changes but neither identify nor discuss the relative roles of greenhouse gases and aerosols toward the warming over the past four decades.

*I have made some specific suggestions below, but overall I would suggest that the authors a) consider changes in exposure and vulnerability in concert with changes in damages when performing trend analysis, and b) thoroughly link rising Mediterranean Sea temperatures to hailstorm activity (specifically, as compared to overall thunderstorm activity) in Europe.*

Thanks, we have addressed both these points in the revised manuscript, as discussed above.

**Specific comments**

*1. Lines 15-25: In the introduction the authors should be more specific – what is meant by "recent times"?*
- "recent times" replaced by "past few years" (now at line 221 of revised manuscript).

*Were the June 2021 losses caused by hail or by flood as hinted at by the authors mentioning saturated soils?*
- the revised text is clearer (lines 221-222): "with 4.5 billion USD of insured losses from severe thunderstorms in Europe during summer 2021"

*What is "similar magnitude" in the case of the Munich storm?*
- this is spelled out in line 226 of the revised text "... since the multi-billion loss in the Munich hailstorm of 1984"

*"A couple of decades" should be replaced by the actual time period examined.*
- this sentence is removed from the revised text

*2. Lines 75-76: The question with insurance loss data is always the proportion of the trend owing to change in the number or type of insured objects (ie the exposure and vulnerability part of the risk). See for example Strader et al. (2024) for tornado risk. The authors should comment on this aspect of uncertainty, and whether they think it plays a big role in the trends they present here.*

The uncertainty in the indexation of older event losses to the present day (or more generally, a common index year) is mentioned in lines 199-200.

Note that the indexation of loss timeseries from national insurance bodies consider both the number and type of risks. In general, the insurance industry has been developing data and methods over a few decades to index their loss experience, because they place much value in knowledge derived from recorded loss data. We hope this message is conveyed clearly in revised text of lines 86-100. A summary of hail trends from the two types of studies (hazard- and loss-based) is given in lines 244-250 of revised text, and it is repeated again in the Conclusions.

*3. Line 89: "Past research points to a warming climate causing more damaging hail in Europe". Yes, but the uncertainty must always be addressed because there remain many unknowns, as the authors have mentioned previously with e.g. references to Raupach et al. (2021). The authors should include the uncertainty in this statement.*

This specific sentence is removed from the revised text.

The revised text provides estimates of rising hail trends as ranges, e.g. lines 244-245 and lines 279-285, to convey uncertainty. It includes a discussion of sources of uncertainty, e.g. lines 196-201, and suggests future research into reducing the uncertainty in the Conclusions (lines 281-282). We also have a new sentence in lines 283-284 of revised manuscript which contains our best understanding of all evidence: "Uncertainty is large, though large-scale trends in the higher-risk area are positive from almost all studies".

*4. Line 125: The Butterworth filter and its use requires a reference.*

A reference to Butterworth (1930) has been added.

*5. Line 187: "rising trend in the European hail climate" – this needs more specificity, since observed trends depend heavily on whether frequency or severity is considered, the geographical region, whether only hail of certain sizes is considered, etc.*

This phrase is from a sentence which contained more specific details – the rising trend was "from hailpad observations, insured losses, and raw weather ingredients for large hail". Nevertheless, the sentence was removed from the revised manuscript, and lines 273-284 in the Conclusions section contains more specific details on the rising trends.

*6. Line 194: "Rising temperatures of local seas humidify the low-level air, which intensifies thunderstorms leading to more severe hail". The key word here is "local". How much effect would rising Mediterranean temperatures have on hailstorms far inland in Europe? This kind of link needs to be much further explored in this article.*

The intention of the article is to investigate Europe-wide behaviour, hence the focus on the higher-risk areas which contribute the most to Europe-wide risk. To avoid confusion, we have reviewed the entire manuscript to clarify our intention to study large-scale behaviour over the higher risk areas.

The revised manuscript also includes discussions of how the Mediterranean affects these high-risk areas (e.g. lines 31-34 of Introduction, lines 171-180 of Section 4, lines 273-275 of Conclusions).

A more detailed study to estimate the impact of Mediterranean warming on hail trends by location is beyond the scope of this article, though we do highlight this as a potential topic for future work, in lines 190-192, and in the final paragraph of the Conclusions.

*7. Line ~200: While aerosol changes have temperature effects, including on the Mediterranean as discussed here, they also have separate effects on thunderstorm activity that are highly uncertain and may exacerbate or modulate the changes owing to increased Mediterranean Sea temperatures. The authors should comment on this.*

Various research has identified the cause of the recent trends in hailstorm activity as being mainly due to increases in low-level moisture, and these are referenced in the text (Kunz et al. 2009; Mohr and Kunz, 2013; Rädler et al., 2019; Battaglioli et al., 2023; Wilhelm et al., 2024). This explanation of hailstorm trends has become well established. Given the length constraints of a Brief Communication, we cannot extend the scope to discuss possible higher-order effects, such as trends in aerosol concentrations and their size distributions.

**Technical corrections and typos, etc**

*1. Line 42: Remove repeated "more recently".*
This paragraph was re-written and does not contain two consecutive sentences beginning 'More recently…'.

*2. Line 83: Suggest removal of "more accurate" since CAPE and wet-bulb temperatures measure different things.*
All mention of wet-bulb temperatures is removed from the revised manuscript.

**References**

Ali, E., W. Cramer, J. Carnicer, E. Georgopoulou, N. Hilmi, G. L. Cozannet, and P. Lionello, 2022: Mediterranean region. Climate Change 2022: Impacts, Adaptation and Vulnerability. Contribution of Working Group II to the Sixth Assessment Report of the Intergovernmental Panel on Climate Change, H. O. P̈ortner, D. C. Roberts, M. Tignor, E. S. Poloczanska, K. Mintenbeck, A. Alegr´ıa, M. Craig, S. Langsdorf, S. L̈oschke, V. M̈oller, A. Okem, and B. Rama, Eds., Cambridge University Press, Cambridge, UK and New York, NY, USA, doi: 10.1017/9781009325844.021.

Raupach, T. H., and Coauthors, 2021: The effects of climate change on hailstorms. Nat Rev Earth Environ, 2 (3), 213–226, doi:10.1038/s43017-020-00133-9.

Strader, S. M., V. A. Gensini, W. S. Ashley, and A. N. Wagner, 2024: Changes in tornado risk and societal vulnerability leading to greater tornado impact potential. npj Natural Hazards, 1 (1), 20, doi:10.1038/s44304-024-00019-6, URL https://doi.org/10.1038/s44304-024-00019-6.

---

## Author Comment (AC2)

**Reviewer 2**

*The manuscript discusses how the risk for damage from hail has changed in the past decades in Europe. It addresses a problem that is not widely studied and is well in the scope of the journal. To showcase that damage from hail has increased, the authors present results from hazard-based studies as well as studies and catalogues looking at insured losses. To study the causes behind this trend, the authors use DAMIP models from CMIP6 to study the effect of different forcing mechanisms on northern Mediterranean sea temperatures which have been shown to be linked with hail-producing storms. They find that sea temperature in the Mediterranean has had a mostly linear response to forcings. They show that from 1850 until 1970 the cooling effect from anthropogenic aerosols dominated the overall trend of Mediterranean temperatures, after which warming effect due to increased greenhouse gases took over and has caused rapid warming since, leading to an increased hail risk in Europe.*

We are very grateful to Reviewer 2 for providing comments which improve the manuscript significantly.

**General comments:**

*Overall, the manuscript is well written and structured clearly .The literature review is conducted thoroughly and the synthesis in the introduction nicely shows how damage from hail has increased in recent decades in parts of Europe. The DAMIP models are cleverly used to attribute different trends in temperatures to different forcings. While this analysis is valid, it could be extended to better substantiate its link to hailstorm damage or risk. I have two main points to highlight:*

1. *While the link between certain variables and the occurrence of hailstorms is established in previous studies, the analysis presented in the manuscript using DAMIP models does not convincingly address the trends in hailstorm damage directly or through these links. To better justify the claims made in the manuscript, the association of surface temperatures (sea or near-surface) to the variables mentioned in the introduction, such as CAPE and near-surface moisture or wet-bulb temperature, should be explained more in detail or demonstrated in the analysis. This would highlight more in accordance with previous literature how hailstorm damage trends are linked with trends of other variables.*

The revised manuscript contains a more detailed explanation of how Mediterranean temperatures are linked to hail damage. An overview is given in the Introduction: how warming waters humidify the air (largely based on your comment 5 below) in lines 26 to 28, then followed by the association of Mediterranean Sea temperatures with hailstorms in lines 30 to 33. Section 4 of the revised text gives fuller details on the connection from Mediterranean moisture to hail trends, in lines 171-193, including several references to past studies showing a direct influence of the Mediterranean moistening low-level air in past major hailstorms. The revised manuscript contains much more evidence of the established link from Mediterranean temperatures to hail damage.

2. *The extent of the analysed area is quite small to claim that conclusions can be drawn for the whole of Europe.*

   We agree that the Mediterranean area in the original analysis was quite small, and now study an expanded Mediterranean area depicted in Figure 1 of the revised manuscript.

   *The mechanism with which conditions in northern Mediterranean affect hailstorm occurrence elsewhere in Europe should be highlighted more than through the mention of a reference.*

   The weather conditions conducive to large hail over mainland Europe are summarised in lines 174-177 of the revised manuscript (specifically, a trough to the west of where hailstorms develop, and how this trough acts to draw Mediterranean air northwards, as described in Kunz et al., 2020).

   *In addition or alternatively, some statistics from previous studies of how often a hailstorm (elsewhere) in Europe is influenced by high dew points in the Mediterranean could be included.*

   The revised manuscript contains our best knowledge of the statistics, in the paragraph at lines 171-180. There are no comprehensive statistics, but Kunz et al. (2020) describe the Mediterranean moisture source in their typical hailstorm case in western Europe, and we also cite research into very severe events in recent decades, which all drew moisture from the Mediterranean.

The revised text also discusses how hail trends at all locations depends on other factors, in addition to Mediterranean warming, in Section 4 (lines 190-192), and in the final paragraph of the Conclusions.

*Moreover, the claim that hailstorm damage can be assessed for whole Europe in this way, is valid only if a significant majority of damage from hail occurs in the mentioned areas in central and western Europe. If this is the case, it should be somehow shown in the manuscript. Otherwise, I suggest the title (and elsewhere in the text) to be more focused on a specific area of Europe.*

Lines 171-180 of the revised manuscript provide the evidence that the Mediterranean has a major influence on severe hail in the higher risk areas of central and southern Europe. Specifically, five references are provided containing evidence of how the Mediterranean is a key source of low-level moisture for damaging hail in the higher risk areas of Europe. We have revised the manuscript to clarify the study concerns higher risk parts of Europe, rather than every location of the continent.

**Specific comments/questions:**

1. *Line 13: Last sentence in the abstract is not factually incorrect but could be rephrased to make it not sound like it is the trends that are warming seas, rather than the forcing. For example, "Given the current trends in anthropogenic forcing, seas are expected to continue warming..."*

The final sentence of the Abstract has been rephrased to clarify how Anthropogenic forcings are warming the Mediterranean.

2. *Line 54: During which time period does the increase of 1 to 1.5% p.a. occur?*

This is clarified in lines 215-216 of the revised manuscript. Your question highlights the different time periods over which trends are computed by various studies. An extra sentence has been added on how the Rädler et al. (2018) trends refer to a time period which is most closely aligned with the Mediterranean warming beginning around 1980.

3. *Line 59: Does Figure 1a show annual losses for automobiles for combined wind and hail perils or only hail, which comprise the vast majority?*

The auto losses are for the combined wind+hail perils. This is clarified in the revised manuscript, lines 236-237.

4. *Line 64: I am unfortunately not very familiar with insurance terminology. How does loss cost relate to total loss? Does it increase with increasing losses?*

The text was revised to define loss cost and provide additional explanation, in lines 96-100 of new subsection 2.2.

5. *Line 79 and 87: Previous studies have found rising low-level humidity and wet-bulb temperature to be linked with an increased hail risk. Is the thermodynamic effect here that SST and near-surface air increase as much, which through increased evaporation retains relative humidity values? Thus wet-bulb temperature rises along with specific humidity?*

Yes, thanks, we include this detailed explanation in the revised text, lines 27-29.

We would add one minor point here, that the wet-bulb temperature depends on air temperature as well as specific humidity, as shown in a psychrometric chart. This means the wet-bulb temperature is not simply a measure of air humidity, though it remains quite widely used by researchers because it is well observed. Note that the revised manuscript makes no reference to wet-bulb temperatures.

6. *Line 108: Although a link to CMIP documentation is provided, Table 1 could include more details about the models used, for example model resolution.*

Three extra columns of information per model have been added to Table 1 in the revised manuscript, including the resolution of atmosphere and ocean models.

7. *Line 111: What is the effect of using near-surface temperature from the model but sea surface temperature from observations? Are SSTs prescribed in the DAMIP models? Is the effect inconsequential because only anomalies are considered?*

The effect is inconsequential because anomalies of SST and near-surface air temperature are similar for the large spatial region (Figure 1) and multidecadal scales analysed here, considering their coupling via heat fluxes. They are often used interchangeably in long historical datasets (e.g. Morice et al., 2012; https://doi.org/10.1029/2011JD017187). More specifically, Rubino et al. (2020; https://doi.org/10.1038/s41598-020-64167-1) showed their anomalies were very similar at large space and long timescales (their Figure 4).

Coupled climate models are used in DAMIP experiments: their SSTs are simulated by the ocean model.

8. *Line 113: What is the purpose of using data from different time periods for observed and modelled values? Is it to highlight the small anomalies at the start of the period and on the other hand the increased anomaly projected to continue after the end of modelled data? I realise that methodologically it has little influence since the anomalies are calculated for the common period.*

Yes, model data from 1850-1869 highlight small impacts from external forcings simulated by DAMIP models in the early period, while the observed data after 2014 show the continued warming, especially in the 2020s. The graphic can provide this extra information, in addition to clearly showing the observed/model overlap period for validation.

9. *Line 120: What are the boundaries of this region based on? Are the modelled near-surface temperatures considered only over the shaded region in Figure 2, i.e. over land? This could be specified in the figure caption or text.*

The revised manuscript analyses a slightly larger region shown in the new Figure 1. The shaded region denotes the area of the Mediterranean Sea considered in this analysis, and this is added to the revised figure caption. Its definition is based on the area of the Mediterranean adjacent to Europe.

10. *Line 139: What is the cause of the reduced amplitude of the multidecadal oscillation? Is due to smoothing from using a multi-model and ensemble mean?*

The answer depends on the relative contributions from internal climate variability and external forcing toward the observed multidecadal oscillation. There are two main possibilities:

   i.  There was a substantial contribution from internal climate variability. In this case, one would expect the DAMIP multimodel mean to simulate a reduced amplitude of this oscillation, since model ensemble members are initialised with different phases of internal variability which would tend to cancel in the multimodel mean.

   ii. Relatively little contribution from internal climate variability. In this case, the reduced amplitude of the multidecadal oscillation from DAMIP models may be due to the external forcings applied to the models not specified accurately, or if the simulation of the climate impact from the forcing is not accurate. There is evidence to suggest this case is plausible: the cooler period starts in the late 19[th] century and persists during a period with three major climate-changing eruptions [Krakatoa in 1883; Santa Maria in 1902, Novarupta in 1912], then the regions warms during the subsequent quieter volcanic period. Models are known to be sensitive to the specification of the volcanic forcing (e.g. Toohey et al., 2014), and it is also known that model simulations of the climate impacts of volcanoes have weaknesses (e.g. Driscoll et al., 2012; Zanchettin et al., 2022). We have no evidence that this multidecadal oscillation is caused by volcanic activity, and merely note it is possible, though the reviewer's suggestion of smoothing from using the ensemble mean is a simpler explanation.)

To our knowledge, there have been no published results on the relative roles of internal climate variability and external forcing toward the observed multidecadal oscillation, and it's not clear if the simpler explanation (i.e. the use of the multimodel mean) is correct.

The above level of detail is too much for our intended Brief Communication, and would distract attention away from the focus on the recent exceptional warming. It is an interesting topic for a future study.

11. Line 143: Does this refer to statistical significance in the amplitude of the multidecadal oscillation in Hist? If so, based on which test?

Yes, the model Control simulations (no external forcing, internal variability only) indicate that internal variability produces multidecadal oscillations with expected amplitude of 0.046 K for a 70-member ensemble mean. In sharp contrast, the simulated amplitude of the oscillation in the Hist ensemble mean is about 0.5 K. Therefore, the signal in Hist is many times larger than the standard deviation due to internal climate variability.

12. *Line 169: Are the more references to substantiate the claim about anthropogenic aerosols driving the multidecadal changes such as the profound peak in European windstorm damages? I would be interested to see the given reference but it is yet to be published.*

The contribution of anthropogenic aerosols (AA) to the historic peak in storminess in the late 20$^{th}$ century is specifically discussed by Cusack and Cox (2025). Hassan et al. (2021) give a more general description of the effects of AA on mid-latitude winds. Note that this point about AA forcing past multidecadal variations in other perils is removed from the revised manuscript.

13. *Line 183: Given the relatively good agreement between models, can you comment on why especially in the most disagreeing models (4 and 6) it looks like the deviation from the multi-model mean increases with time in Figure 3b? Is this an effect of the multi-model mean diverging from the climatological mean, i.e. larger forcing introducing more spread between the models?*

It appears that their relative deviations (i.e. anomaly / multimodel mean signal) do not grow in time. Climate models can contain different sensitivities to applied forcings, and as these forcings increase in time, these different sensitivities will lead to larger differences in impacts.

It is interesting to note how model 4 has the largest sensitivity to both GHG and Aero, while model 6 has the smallest sensitivity to both. It would require expert analysis to gain insight into why different models produce different impacts on the Mediterranean Sea temperatures for a given external forcing, and it is not clear whether the necessary diagnostics are available. Such an investigation is beyond the scope of the study.

14. *Line 199: Although the overall trend between 1850 and 1970 is a cooling one, to my ear the way this sentence is phrased makes it sound like there was continuous cooling throughout the period.*

The phrasing has been adjusted to "The Aero effect dominated up to the late 1970s to produce a slightly cool period on average,…" in lines 267-268 of the revised manuscript.

**Technical comments/typos:**

1. *Line 45: I believe here and elsewhere "Radler" in citation should be "Rädler".*
Thanks, this has been corrected throughout the manuscript.

2. *Line 142: Symbol for standard deviation does not show properly.*
Thanks, it's unclear what went wrong in original word document, and it has been fixed in the revised text.

3. *Line 158: The word "then" is repeated in the sentence.*
Fixed in revised manuscript.

**References used in replies to Reviewer 2**

Driscoll, S., A. Bozzo, L. J. Gray, A. Robock, and G. Stenchikov 2012: Coupled Model Intercomparison Project 5 (CMIP5) simulations of climate following volcanic eruptions, J. Geophys. Res., 117, D17105, https://doi.org/10.1029/2012JD017607.

Hassan, T., R. J. Allen, W. Liu, and C. A. Randles, 2021: Anthropogenic aerosol forcing of the Atlantic meridional overturning circulation and the associated mechanisms in CMIP6 models. Atmos. Chem. Phys., 21, 5821–5846, https://doi.org/10.5194/acp-21-5821-2021.

Toohey M., Krüger K., Bittner M., Timmreck C., Schmidt H. 2014: The impact of volcanic aerosol on the Northern Hemisphere stratospheric polar vortex: mechanisms and sensitivity to forcing structure, Atmos. Chem. Phys., 14, 13063–13079, https://doi.org/10.5194/acp-14-13063-2014.

Zanchettin D., Timmreck C., Khodri M., Schmidt A., Toohey M., et al. 2022: Effects of forcing differences and initial conditions on inter-model agreement in the VolMIP volc-pinatubo-full experiment, Geosci. Model Dev., 15, 2265–2292, https://doi.org/10.5194/gmd-15-2265-2022.

---

## Author Comment (AC3)

**Reviewer 1**

*This manuscript aims to assess potential trends in hailstorm damage across Europe using climate modelling. The study aligns well with the scope of the journal, addressing a topic that is both timely and underexplored in previous studies. However, in my opinion the manuscript suffers from a notable methodological limitation: the main approach and results only loosely correspond to the stated objective of the study.*

*Specifically, the authors utilize modelling outcomes from the Detection and Attribution Model Intercomparison Project (DAMIP) to demonstrate that warming sea surface temperatures in the Mediterranean are predominantly attributable to anthropogenic influences. While it is justifiable that rising sea temperatures correlate with increased hailstorm activity, this connection is neither directly analyzed in the manuscript nor explored through more appropriate proxy variables, such as wet-bulb temperature or measures of convective instability. Sections 2 and 3 (Methodology and Results) bear no direct relationship to hailstorm trends, apart from the general hypothesis that continued warming will lead to an increase in hail-related damage.*

*While the current methodology requires refinement to align with the study's objectives, the authors have demonstrated a significant effort in leveraging DAMIP outcomes to explore anthropogenic influences on sea surface temperatures. Thus, despite my overall concerns, I am not suggesting rejection. Instead, I encourage the authors to undertake a major revision to address the above-mentioned issues, acknowledging that this will involve significant challenges. One possible path forward would be to incorporate more targeted hail-related variables into the methodology. Alternatively, the authors could expand their literature review on hail damage, employing more rigorous analytical techniques and relocating this analysis from the Introduction to the Methodology and Results sections.*

Comments from reviewer 1 led to large improvements in the revised manuscript, and were much appreciated by the authors.

We agree with the main comment from Reviewer 1. The original manuscript was designed as a Brief Communication, to contain a short analysis of the forcing of Mediterranean Sea temperatures, and how its ongoing warming has consequences for hail damages in Europe. However, the original Title, and other statements in the main text, suggested the study concerned hailstorms, and we apologise for creating confusion and misleading readers into expecting a study centred on European hailstorms.

The revised text is intended to clarify the aims of this study, and the analysis performed. It includes a full re-framing of the aims of the study in the Abstract and Introduction, and the manuscript has been re-structured with new sections 3 and 4, to make the distinction between the warming Mediterranean (Sect. 3), and consequences on severe weather (Sect. 4). A more detailed review of how hail damage is connected with the Mediterranean is also given in the new Section 4. This structure and content is aligned with the reviewer's suggested alternative path forward.

**Specific & Minor Comments:**

*1) Lines 18 – 19: Please verify whether this is the appropriate format for referencing a website according to the journal's guidelines.*

Thanks, all references to websites in the revised text have been modified to meet journal guidelines.

*2) Some abbreviations appear unnecessary or unconventional, such as "R21" for Raupack et al. (2021), and "the Med" for the Mediterranean Sea.*

These abbreviations have been removed.

*3) Figure 1. It is unclear whether this figure represents the authors' original analysis or a direct visualization of data from other studies. If it is original, it should be moved to the Results section. If it is based on external data, the appropriate references must be included in the figure caption.*

This figure has become Figure 4 in the new Section 4 of the revised text, and external data are described in section 2, and referenced in the new caption.

*4) Lines 74 – 77. The above comment regarding clarification of data sources applies here as well.*

This text is relocated to lines 244-245 in the revised manuscript, and the data sources for both the damage-based and hazard-based estimates are clarified.

*5) Lines 106 – 107. Please specify which monthly-mean near-surface temperature diagnostics were used in this study to ensure transparency.*

The CMIP variable name 'tas' has been added to the revised text (line 54).

*6) Consider expanding Table 1 to include more detailed information about the modeling simulations, such as specific parameters, assumptions, or configurations.*

Three new columns of information have been added to Table 1: the vintage of the model, and the resolutions of their atmosphere and ocean components.

*7) Lines 152 – 153. The information about the red tick marks should also be included in the figure caption for clarity.*

The figure caption has been revised.

---

## Author Comment (AC4)

**Community Comment by Cameron Rye**

*I have a comment about the trends in insured losses reported in the paper. Further details around the methodology used to adjust historical losses to present-day values is required. For example, it appears that GDV losses for Germany automobiles have been adjusted for inventory (number of cars) and inflation. However, the standard method for adjusting insured losses is to account for three factors: inflation, wealth and inventory. I suspect wealth has not been taken into account in this instance. The $ value of cars has increased significantly over time, particularly with the introduction of electric vehicles. As a result, I suspect the average pay-out for a auto hail claim will have significantly increased over time. How can the authors be sure that the trend in Figure 1a is due to increasing risk and not simply reflecting the increasing value of vehicles? The same point applies to other datasets, for example it appears that the France property data have only been adjust for the cost of reconstruction (FFB index), and not other factors.*

Thanks, your comment has highlighted how the description of indexation in the original manuscript was lacking information. The text has been revised to include more details on the indexation of losses by national insurance associations, in lines 86-100, to highlight the sophisticated indexation techniques used in insurance.

Regarding Germany data, the GDV kindly responded to our request for more information on how they trend vehicle losses: they described how their indexation takes account of the cost of parts and repair, and the number of vehicles. (Also note how electric vehicles are a small fraction of the market, e.g. from Statista: "As of 2024, electric vehicles still had a low market share in Germany, at around three percent for battery-powered electric vehicles and almost two percent for plug-in hybrids." See https://www.statista.com/statistics/1166826/electric-vehicles-market-share-germany/.)

The adjustment for France includes the growth in the number of insured risks, in addition to the growth in the reconstruction cost. This is described in more detail in the revised text. There is one aspect of growth in wealth which is not covered, regarding how part of our growing wealth is manifested as a greater value of contents inside the insured structures. However, the vast majority of hail claims concern damage to the building envelope (i.e. roof, siding, windows, doors) rather than damage to contents.

Information from VKG in Switzerland consist of loss cost data, in which total claims are expressed as a ratio of the total insured value for each year. This means their indexation relies on how total insured value is defined each year. In the insurance industry, the total insured value is designed to capture those factors causing growth in claims.

*Secondly,*

- *I suggest that the focus of the paper could be expanded to consider other perils. The med ocean is also important for e.g. medicines, cut-off lows, Vb cyclones.*

We agree that the Mediterranean warming has implications for other perils, such as different types of flood events. The flood peril is now mentioned in the revised text, though we do not conduct an in-depth study because the article is designed as a Brief Communication rather than a full article, hence has been limited to focus on hail. Research into how the warming Mediterranean may have affected flood severity would be an interesting piece of future work.

- *Hail risk is complex, and not entirely dependent on the med ocean (i.e. there is not necessarily a direct 1-to-1 link).*

The Brief Communication concerns trends in the hail risk, and various research has identified the cause of the recent trends in hailstorm activity as being mainly due to increases in low-level moisture (Kunz et al. 2009; Mohr and Kunz, 2013; Púčik et al., 2017; Taszarek et al., 2021; Wilhelm et al., 2024). Further, the Mediterranean Sea is the main source of low-level moisture for damaging hailstorms in the higher-risk parts of Europe, as discussed more fully in the revised text (e.g. lines 31-34, 171-180, and several references therein).

- *In my view it would be a more rounded article if the focus is the med, and then talk about how this is important for a number of perils.*

A review of the impact of the warming Mediterranean on multiple perils – in addition to analysing the drivers of the warming Mediterranean – is beyond the scope of a Brief Communication. Though we agree it would be an interesting subject for future work.

---

## Referee Report (RR1)

**Review of NHESS-2024-210**

Re-review of NHESS-2024-210. I thank the authors for their responses to my comments from my first review and for the significantly refocused and much improved article.

In my last review I suggested that the authors a) consider changes in exposure and vulnerability in concert with changes in damages when performing trend analysis, and b) thoroughly link rising Mediterranean Sea temperatures to hailstorm activity (specifically, as compared to overall thunderstorm activity) in Europe. The authors have better explained how their loss estimates are normalised for exposure and vulnerability changes, which is helpful for part a). For part b), the authors have relied on a review of previous studies that show the importance of low-level moisture for convection-prone (not always hail) environments. The importance of low-level moisture is now well established, but the link from rising Mediterranean temperatures to severe hail trends is still not thoroughly made.

The secondary aim of the paper is to "quantify the link from Mediterranean Sea warming to its impacts on hailstorm risk in Europe". The authors have not succeeded in this aim. There is no quantification in the paper; rather the two trends are analysed in parallel. At a minimum, I suggest that the authors quantify the correlation between Mediterranean temperatures and hail damage or hail-prone environments in the historical period. Do years in which the Mediterranean is warmer produce more severe hail? How much of the variability in hail losses is (statistically) explained by variability in Mediterranean temperatures? Plotting the two timeseries together would also help show a relationship.

Secondly, the authors should take great care in their wording of conclusion statements so that the uncertainties inherent in this study are well explained – I show specifics below. While I do not doubt that the warming Mediterranean plays a role in the trends in severe hail in Europe, this study does not yet show a convincing link. I hope that my comments below will help to improve the manuscript.

**Specific comments**

1. Lines 32: The cited article by Kunz et al. (2018) does not show a direct link between Mediterranean moisture and hailstorms.

2. Line 54: The authors analyse the variable "tas", or near-surface atmospheric temperature,

rather than model variables for sea surface temperature which may be different. They compare these data to historical sea surface temperature in Figure 3. The authors should mention this difference and explain their choice. In Figure 3, the agreement of the historical simulations with observations is not particularly strong, with observations showing much more variability. The authors should comment on this discrepancy in variability.

3. Line 181: The authors write "A number of studies identify increases in low-level moisture as being the main cause of rising hailstorm risk across Europe over the past few decades" – but no references are given. The authors should cite exactly which studies show this, because not all the studies they cite in the following paragraph show that low-level moisture is the main driver of increases in hailstorm risk – rather some show the importance of low-level moisture in more-general convective environments, of which hail environments are a specific subset.

4. Line 191: The authors write that "it is clear that a warming Mediterranean is a primary contributor to the trends in hail risk in key parts of Europe". The authors have shown that moisture increases are often linked to increases in convective storm environments, but the link to trends in hail risk is not clear. Other factors such as changes in melting of hailstones and local changes in convective inhibition may affect hail hazard, while risk changes are also affected by changes in vulnerability and exposure. I would suggest allowing for more uncertainty by replacing this line with "At the present time, it is likely that a warming Mediterranean is a primary contributor to the trends in the occurrence of convective environments in some parts of Europe".

5. Around line 215: The authors use "hail risk" when they may be referring to hail hazard. I suggest only using the word "risk" when exposure and vulnerability are also taken into account in these reported results.

6. Line 238: "both trends are significantly different from zero at the 1% level" and similar lines – which statistical test is used for significance statements?

7. Line 275: "are driving hail trends since about 1980" – there is not sufficient evidence for this claim. Trends in hail are highly uncertain and complex, with large geographical inhomogeneity, and there is not one single driver.

8. Lines 275–278: "Recent trends in hail damages over these higher-risk parts of Europe were reviewed to measure the impacts due to Mediterranean warming." This study does not measure the impact on hail due to Mediterranean warming – there is no quantification. Rather, the study shows that a) the Mediterranean has warmed, b) low-level moisture from the Mediterranean is important for convection, c) other published severe hail trends show increases over Europe, and d) hail damages have also increased over Europe. The link between a) and c) and d) needs to be made stronger and it is not shown whether the Mediterranean temperatures are the key driver amongst many influences on severe hail production.

**References**

Kunz, M., U. Blahak, J. Handwerker, M. Schmidberger, H. J. Punge, S. Mohr, E. Fluck, and K. M. Bedka, 2018: The severe hailstorm in southwest germany on 28 july 2013: characteristics, impacts and meteorological conditions. *Quarterly Journal of the Royal Meteorological Society*, **144 (710)**, 231–250, doi:10.1002/qj.3197, URL https://rmets.onlinelibrary.wiley.com/doi/full/10.1002/qj.3197.

---

## Author Response (AR2)

**Reviewer 2**

*I suspect the sentence on line 188 is missing an object.*

Thanks for finding this error. This sentence was missing a word and is corrected in the new version of the manuscript.

**Reviewer 3**

*Re-review of NHESS-2024-210. I thank the authors for their responses to my comments from my first review and for the significantly refocused and much improved article.*

The authors thank Reviewer 3 for their time and effort on this manuscript. Their comments highlight aspects that may cause confusion, and we appreciate this opportunity to revise and clarify details in the manuscript.

*In my last review I suggested that the authors a) consider changes in exposure and vulnerability in concert with changes in damages when performing trend analysis, and b) thoroughly link rising Mediterranean Sea temperatures to hailstorm activity (specifically, as compared to overall thunderstorm activity) in Europe. The authors have better explained how their loss estimates are normalised for exposure and vulnerability changes, which is helpful for part a). For part b), the authors have relied on a review of previous studies that show the importance of low-level moisture for convection-prone (not always hail) environments. The importance of low-level moisture is now well established, but the link from rising Mediterranean temperatures to severe hail trends is still not thoroughly made.*

The connection from Mediterranean moisture to rising hail trend is queried multiple times by the reviewer, so we revised the manuscript to highlight the evidence on this important topic. Section 4 contained a lot of information on multiple subjects, and quite monolithic, and has been re-structured into two subsections, with modified text, in the new version of the manuscript. In particular, subsection 4.1 contains a review of studies showing the Mediterranean is the main source region for the high values of low-level moisture that are a key ingredient of damaging hail events, then reviews past research which found that rising amounts of low-level moisture were the main cause of upward trends in damaging hail. The revised manuscript presents the published evidence connecting Mediterranean heat to damaging hail in key parts of Europe more clearly to the reader.

*The secondary aim of the paper is to "quantify the link from Mediterranean Sea warming to its impacts on hailstorm risk in Europe". The authors have not succeeded in this aim. There is no quantification in the paper; rather the two trends are analysed in parallel. At a minimum, I suggest that the authors quantify the correlation between Mediterranean temperatures and hail damage or hail-prone environments in the historical period. Do years in which the Mediterranean is warmer produce more severe hail? How much of the variability in hail losses is (statistically) explained by variability in Mediterranean temperatures? Plotting the two timeseries together would also help show a relationship.*

The study consists of three parts:

1. Identifying the drivers of multidecadal changes in Mediterranean Sea temperatures over past decades
2. Review of past studies establishing how recent trends in Mediterranean Sea temperatures drive trends in damaging hail over higher-risk parts of western and central Europe
3. Reviewing trends in damaging hail, including new analysis of losses from national insurance associations

Parts 2 and 3 above were described inaccurately in the manuscript, and we thank the reviewer for identifying the weakness in the sentence they highlight. We replaced this sentence with the following text in the revised version: "Section 4 begins with a review of the robust evidence linking multidecadal trends in Mediterranean Sea temperatures to damaging hail over the higher-risk parts of Europe, then we provide estimates of trends in damaging hail over recent decades, based on previous studies and a new analysis of insurance loss data."

Note how the study establishes the connection from Mediterranean warming to damaging hail in high-risk parts of Europe using past research findings. It is not possible to precisely quantify such a link due to limitations of hail damage records (as noted in first sentence of subsection 4.1), but past research has established a strong connection. The contributions of this Brief Communication are to identify the drivers of the Mediterranean warming, and to review trends in damaging hail from other studies, and loss datasets, and we use the findings from past research to highlight how both trends are connected.

We considered the request from the reviewer to analyse the correlation between Mediterranean heat and hail at annual timescales. This study concerns the connection of their multidecadal trends rather than interannual variability. There are substantial differences between behaviours at these two timescales, which means correlations at annual timescales do not reflect the strength of connection at multidecadal scales. For example, damaging hail is rare and the sampling error at annual scales naturally reduces correlation, whereas a multidecadal analysis has much

smaller sampling error. Further, the interannual variations of other hailstorm ingredients will reduce correlation between Mediterranean heating and damaging hail at annual timescales, but researchers have found these other ingredients have very little multidecadal trend. Therefore, the inter-annual correlation has little relevance to a study of multidecadal trends. Note that the manuscript provides the timeseries of multidecadal trends in Mediterranean temperatures and hail losses: the red solid line in Figure 2, versus the red dashed lines in Figures 4a-c.

*Secondly, the authors should take great care in their wording of conclusion statements so that the uncertainties inherent in this study are well explained – I show specifics below. While I do not doubt that the warming Mediterranean plays a role in the trends in severe hail in Europe, this study does not yet show a convincing link. I hope that my comments below will help to improve the manuscript.*

Your comments have improved the clarity and accuracy of the manuscript, and the authors are very grateful.

We reviewed the conclusions in section 5 to ensure they accurately reflect uncertainties in the coupling between Mediterranean Sea temperatures and damaging hail occurrence in this higher-risk parts of western and central Europe. Several independent studies conclude that the trends in low-level moisture are the main driver of the trends in damaging hail in high-risk parts of Europe (subsection 4.1). Other studies show how low-level Mediterranean air masses are the typical source of the high moisture values which are key to damaging hail events in the high-risk parts of Europe (subsection 4.1). These studies combine to provide confidence in a strong connection between Mediterranean Sea temperatures and damaging hail occurrence at multidecadal scales. There is some remaining uncertainty, referred to in the first sentence of the final paragraph in section 5, and this would be a good avenue for future work.

**Specific comments**

1. *Lines 32: The cited article by Kunz et al. (2018) does not show a direct link between Mediterranean moisture and hailstorms.*

Subsection 4.1 of Kunz et al. (2018) show the link between Mediterranean moisture and the hailstorm that caused major damage in southwest Germany on 28$^{th}$ July 2013. The authors describe how a low pressure off the east coast of Scotland was connected to a secondary low near the Gulf of Lyon, and their Figure 3 includes surface pressure contours (white lines) indicating low-level airflow from the Mediterranean to southwest Germany.

2. *Line 54: The authors analyse the variable "tas", or near-surface atmospheric temperature, rather than model variables for sea surface temperature which may be different. They compare these data to historical sea surface temperature in Figure 3. The authors should mention this difference and explain their choice.*

This point was raised in the first round of review too. The revised manuscript includes the following sentences in the second paragraph of subsection 2.1:

"These SST data are used to assess the validity of near-surface air temperature variations in DAMIP simulations. Anomalies in both quantities are very similar at the large spatial and temporal scales analysed in this study. For example, Rubino et al. (2020) described their close correspondence at interdecadal timescales (their Figure 4), and the two quantities are often used interchangeably in long climate reconstructions (e.g. Morice et al., 2012). "

*In Figure 3, the agreement of the historical simulations with observations is not particularly strong, with observations showing much more variability. The authors should comment on this discrepancy in variability.*

We assume the reviewer is referring to the interdecadal variations from 1850 to about 1980. Reviewer 2 asked about this in the first round of review, and a detailed reply was provided. The manuscript was modified to include a brief discussion of these smaller, earlier variations. Specifically, the first two paragraphs of subsection 3.2 were replaced with the following text:

"The causes of the more rapid warming of the Mediterranean Sea are now explored using DAMIP modelling results. Figure 3a shows the timeseries of Mediterranean temperature anomalies over the extended historical period for observed and multi-model ensemble means for the Historical and three DAMIP single-forcing experiments. Earlier multidecadal variations in observations, consisting of minima around 1910 and the late 1970s, and a local maximum

from about 1930 to the early 1960s, are replicated in Hist, albeit with reduced amplitude. To our knowledge, there are no published results on the relative roles of internal climate variability and external forcing toward the observed multidecadal variations in the Mediterranean basin prior to 1980. However, these earlier fluctuations are similar to those in the North Atlantic sector analysed by Booth et al. (2012), and those authors found they were caused by both anthropogenic and volcanic aerosol forcing, in addition to internal climate variability. The observed warming since about 1980 is of greater concern since it has four times larger amplitude than the mid-20th century peak. Notably, the multimodel ensemble mean simulates this recent rapid warming accurately. This comparison to observed variations bolsters confidence in the fidelity of CMIP6-DAMIP climate model simulations of surface temperature anomalies in the Mediterranean, particularly its dominant feature of rapid warming in recent decades. We now use the validated model ensemble to assess the contribution of individual forcings to the total signal."

3. *Line 181: The authors write "A number of studies identify increases in low-level moisture as being the main cause of rising hailstorm risk across Europe over the past few decades" – but no references are given. The authors should cite exactly which studies show this, because not all the studies they cite in the following paragraph show that low-level moisture is the main driver of increases in hailstorm risk – rather some show the importance of low-level moisture in more-general convective environments, of which hail environments are a specific subset.*

The reviewer quotes the first sentence to the second paragraph of subsection 4.1. This sentence is intended to introduce the reader to the topic of the link between low-level moisture to damaging hail, and the fact that there are two pathways.  The remainder of the paragraph describes the two pathways, and provides several references to studies which link greater moisture specifically to more damaging hail.

One of the references in the reviewed manuscript (Púčik et al., 2017) concerned convective environments rather than observed hail hazard or losses. While there is much evidence linking their convection proxy (high instability, shear and precipitation) to severe hail, this reference was removed because several other studies are referenced to help avoid doubts that moisture trends are connected to damaging hail trends.

4. *Line 191: The authors write that "it is clear that a warming Mediterranean is a primary contributor to the trends in hail risk in key parts of Europe". The authors have shown that moisture increases are often linked to increases in convective storm environments, but the link to trends in hail risk is not clear. Other factors such as changes in melting of hailstones and local changes in convective inhibition may affect hail hazard, while risk changes are also affected by changes in vulnerability and exposure. I would suggest allowing for more uncertainty by replacing this line with "At the present time, it is likely that a warming Mediterranean is a primary contributor to the trends in the occurrence of convective environments in some parts of Europe".*

The second paragraph in subsection 4 was re-written, for clarity.
- Subsection 4.1 refers to a number of studies showing moisture increases are the main driver of upward trends in *damaging hail* in the high-risk parts of central and western Europe
- The third paragraph in subsection 4.1 discusses how hail hazard events depend on other ingredients, then mentions past investigations which conclude that these ingredients have mostly flat or even slightly inhibiting trends, which strengthens the case for rising moisture driving upward trends in damaging hail
- Past research, and this study, take account of changing exposure when assessing damage trends

As a result, we use a version of the reviewer's suggested sentence which is adapted to fit with the evidence presented in subsection 4.1: "Though it is very likely that a warming Mediterranean is the main contributor to the recent trends in damaging hail across the higher-risk parts of Europe."

5. *Around line 215: The authors use "hail risk" when they may be referring to hail hazard. I suggest only using the word "risk" when exposure and vulnerability are also taken into account in these reported results.*

Thanks, this has been changed to "…both report hazard driving changes in hail risk…".

6. *Line 238: "both trends are significantly different from zero at the 1% level" and similar lines – which statistical test is used for significance statements?*

The revised manuscript has extra text inserted at the first mention of significance testing (in the second last paragraph of subsection 4.2): "…both trends are significantly different from zero at the 1% level based on a standard two-sided t-test (used in all significance testing of trends)."

7.  Line 275: "are driving hail trends since about 1980" – there is not sufficient evidence for this claim. Trends in hail are highly uncertain and complex, with large geographical inhomogeneity...

Trends in damaging hail at local scales are spatially inhomogeneous, mainly due to the relative rarity of damaging hail at a location leading to large sampling error. However, there is a large body of evidence showing robust positive trends when damaging hail occurrences are analysed over aggregate spatial scales. The new subsection 4.2 describes the evidence from past research leading to the conclusion that damaging hail has been trending upward since about 1980 across the main high-risk areas of central and western Europe. In more detail, subsection 4.2 refers to many different hazard-based studies, using a variety of input data and methods, and all find a rising trend in damaging hail in the higher-risk parts of western and central Europe. Further, subsection 4.2 refers to past studies showing an upward trend in insured losses in France and Germany, and this study provides a new analysis of data from the French, German and Swiss insurance associations, and all three countries show positive trends in national hail losses. In summary, there is much evidence of rising trends in damaging hail across the higher-risk parts of western and central mainland Europe.

-   … and there is not one single driver

Past research indicates low-level moisture amounts are the main driver of the multidecadal, large-scale trends in damaging hail in the higher-risk parts of western and central mainland Europe. The research consists of a range of different studies all converging on the conclusion that rising low-level moisture has been the main cause of upward trends in damaging hail in recent decades. These past studies are mentioned in the new subsection 4.1, which also describes studies investigating other ingredients of damaging hail events, such as winds and cap strength, and how these other drivers explain little of the trend in damaging hail occurrence. The final paragraph of subsection 4.1 describes how other drivers may be responsible for some of the rising trend, and that further research would be valuable.

8.  Lines 275–278: "Recent trends in hail damages over these higher-risk parts of Europe were reviewed to measure the impacts due to Mediterranean warming." This study does not measure the impact on hail due to Mediterranean warming – there is no quantification.

Please see our reply in the General Comments section, on the scope of this work:
1.  Identifying the drivers of multidecadal changes in Mediterranean Sea temperatures over past decades
2.  Review of past studies establishing how recent trends in Mediterranean Sea temperatures drive trends in damaging hail over higher-risk parts of western and central Europe
3.  Reviewing trends in damaging hail, including new analysis of losses from national insurance associations

We have re-written the final paragraph of the Introduction for clarity on the scope of this Brief Communication. Specifically, the study uses the results and conclusions from past research to establish the connection from Mediterranean warming to upward trends in damaging hail across the higher-risk parts of central and western Europe. The studies which have established this connection are reviewed in subsection 4.1, and also discussed in reply to the second part of question 7, above.

Given how past research has established the link from Mediterranean warming to damaging hail in the higher-risk parts of Europe, then the synthesis of trends in damaging hail in subsection 4.2 is quantifying the impact on hail due to Mediterranean warming. Though we recommend further analysis of the link from Med warming to damaging hail in two key parts of the manuscript: the final paragraph of subsection 4.1, and the final paragraph of the Conclusions.

Rather, the study shows that a) the Mediterranean has warmed, b) low-level moisture from the Mediterranean is important for convection, c) other published severe hail trends show increases over Europe, and d) hail damages have also increased over Europe.

The revisions to the manuscript may clarify the first two parts of the study: (a) the study investigates the drivers of the Mediterranean warming (as described in the Title, Abstract, Introduction, the entirety of Section 3, and the

Conclusions), and (b) low-level moisture from the Mediterranean is important for damaging hail occurrence rather than general convection. This focus on severe hail is mentioned in the Introduction, and reviewed in section 4, and mentioned again in the Conclusions. We are confident most readers will be able to describe what this study has done with regards to parts (a) and (b), since they are described multiple times in the revised manuscript.

> *The link between a) and c) and d) needs to be made stronger and it is not shown whether the Mediterranean temperatures are the key driver amongst many influences on severe hail production.*

The link from (a) to (c)+(d) is made in the new subsection 4.1 in the revised manuscript. Also, please see our replies in the General Comments section, and to Specific Comments numbers 3, 4, and 7 above, for more details on this same point.